# Analysis of Yes-Associated Protein-1 (YAP1) Target Gene Signature to Predict Progressive Breast Cancer

**DOI:** 10.3390/jcm11071947

**Published:** 2022-03-31

**Authors:** Gomathi Venkatasubramanian, Devaki A. Kelkar, Susmita Mandal, Mohit Kumar Jolly, Madhura Kulkarni

**Affiliations:** 1Centre for Translational Cancer Research: A Joint Initiative of Prashanti Cancer Care Mission and Indian Institute of Science Education and Research, Pune 411016, India; vgomathi98@gmail.com (G.V.); dakelkar@gmail.com (D.A.K.); 2Prashanti Cancer Care Mission, 1-2 Kapil Vastu, Senapati Bapat Road, Pune 411016, India; 3Centre for BioSystems Science and Engineering, Indian Institute of Science, Bangalore 560012, India; susmitam@iisc.ac.in (S.M.); mkjolly@iisc.ac.in (M.K.J.)

**Keywords:** yes-associated protein-1, breast cancer, cancer prognosis, gene signatures

## Abstract

Breast cancers are treated according to the ER/PR or HER2 expression and show better survival outcomes with targeted therapy. Triple-negative breast cancers (TNBCs) with a lack of expression of ER/PR and HER2 are treated with systemic therapy with unpredictable responses and outcomes. It is essential to investigate novel markers to identify targeted therapies for TNBC. One such marker is YAP1, a transcription co-activator protein that shows association with poor prognosis of breast cancer. YAP1 transcriptionally regulates the expression of genes that drive the oncogenic phenotypes. Here, we assess a potential YAP target gene signature to predict a progressive subset of breast tumors from METABRIC and TCGA datasets. YAP1 target genes were shortlisted based on expression correlation and concordance with YAP1 expression and significant association with survival outcomes of patients. Hierarchical clustering was performed for the shortlisted genes. The utility of the clustered genes was assessed by survival analysis to identify a recurring subset. Expression of the shortlisted target genes showed significant association with survival outcomes of HER2-positive and TNBC subset in both datasets. The shortlisted genes were verified using an independent dataset. Further validation using IHC can prove the utility of this potential prognostic signature to identify a recurrent subset of HER2-positive and TNBC subtypes.

## 1. Introduction

According to GLOBOCAN, 2.2 million new incidences of breast cancer cases were reported worldwide in 2020, with more than 50% mortality rate in developing countries [1]. Breast cancer is a heterogeneous disease; the most common and aggressive breast cancer type is called invasive ductal carcinoma (IDC), with an incidence rate of 80% of all breast cancers [2]. IDC is further classified based on the molecular expression of estrogen receptor (ER), progesterone receptor (PR) and human epidermal growth factor receptor-2 (HER2) [3]. Of the IDC cases, 60–75% express ER and are responsive to targeted hormonal therapy, 20–25% express HER2 and are associated with aggressive clinical course and poor survival but are responsive to HER2-targeted treatment [4,5]. The remaining 10–15% do not express ER, PR and HER2 and are referred to as triple-negative breast cancer (TNBC). TNBC generally presents with high grade and lymph node positivity and is associated with a poor prognosis due to a lack of targeted therapies [2]. In African American women and in developing countries such as India, the incidence rates of these particularly aggressive subtypes are reportedly higher, up to 22–25% [6,7].

With the advent of targeted therapy in breast cancer, mortality is mostly attributed to cancer recurrence. The survival rate dramatically declines with an increase in the stage, while the recurrence rate is higher in aggressive cancer types such as TNBC [8]. With the unpredictability of treatment response and higher risk of recurrence, there is a need to develop new strategies to identify prognostic and targetable markers to improve treatment response and survival outcomes in such aggressive diseases.

The hippo signaling pathway has been recently shown to be involved in aggressive solid tumor progression [9]. YAP1 and TAZ are downstream effectors of the Hippo pathway and are often activated in many human malignancies [10]. Dysregulation of the Hippo pathway and its downstream effector protein YAP1 in tumor samples could be prognostic markers to predict progressive cancers. Further, these effector proteins could be used to develop targeted therapies to inhibit cancer progression.

A comprehensive molecular characterization of the Hippo signaling pathway across 33,000 sample sets of various cancers showed that high YAP1 expression is associated with poor survival across various cancer types [11]. YAP1 is a transcriptional co-activator that interacts with TEADs (transcriptional enhancer factor domain family) and exerts diverse effects on tumorigenesis and cancer progression [12]. Sustained nuclear localization of YAP1 and transcriptional activation are associated with tumorigenesis and metastatic progression [13]. YAP1 induces the expression of genes related to cell proliferation and indirectly controls the cell cycle by activation of other proto-oncogenic transcription factors such as c-Myc [14]. YAP1 also contributes to the survival of cancer cells by inhibiting apoptosis-associated genes. YAP1 regulates EMT- and stem-cell-associated gene expression, which contributes to tumor progression, resistance to chemotherapy and metastasis [10,15]. Thus, YAP1 plays an important role in tumor progression by regulating the expression of target genes involved in proliferation, EMT, stemness and chemoresistance.

YAP1 overexpression is associated with poor survival in multiple cancer types [16,17,18,19,20]. YAP1 promotes cell proliferation and inhibits apoptosis of breast cancer cells [21]. YAP1 is shown to mediate tumor-promoting functions such as stemness and epithelial-to-mesenchymal transition in mammary epithelial cell lines [22,23,24]. Inhibition of YAP1 has improved response to radiation therapy in TNBC cell lines [25].

YAP1 is a transcriptional co-activator protein regulated by the Hippo signaling pathway [26]. Loss of Hippo signaling leads to nuclear translocation of YAP1 from the cytoplasm and transcriptional activation of genes involved in cell proliferation, EMT and stemness [27,28]. YAP1 target genes have been implicated to play a role in tumor progression. For example, THBS1 (thrombospondin 1) activates focal adhesion kinase when transcriptionally activated by YAP1, which regulates breast cancer progression and metastasis, promoting tumor aggressiveness [29]. Another YAP1 target gene, COL12A1 (collagen type 12 alpha one chain), has been shown to play a role in tumor invasion and migration in gastric, colorectal and breast cancer [30,31]. With YAP1 established as an oncogene in breast cancer, multiple studies have been conducted to assess the mechanistic role of YAP1 target genes in promoting oncogenic phenotypes in mammary epithelial cell lines and breast cancer cell lines [22,32]. The potential role of YAP1 and its target genes in promoting tumor progression in clinical samples needs to be assessed. Detailed association studies of YAP1 target genes with breast cancer outcomes can further improve our understanding of the contribution of YAP1 to breast cancer progression and identify novel prognostic factors.

Earlier, we analyzed the YAP1 target gene signature derived from YAP1-expressing stable mammary epithelial cell lines and showed an association with survival outcomes of breast cancer patients from publicly available datasets [22]. In this report, we analyze the publicly available breast cancer gene expression datasets (TCGA and METABRIC) to evaluate the association of these signature genes individually and in combination with YAP1. The target genes that showed significant association with the outcomes were shortlisted as prognostic candidates to identify a progressive subset of breast cancer.

## 2. Materials and Methods

### 2.1. Breast Cancer Dataset Selection

Publicly available breast cancer datasets, METABRIC and TCGA–BRCA, were accessed for YAP1 and target gene expression data and associated clinical and patient follow-up data. METABRIC (Molecular Taxonomy of Breast Cancer International Consortium) is a collection of 2506 clinically annotated primary fresh-frozen breast cancer specimens from tumor banks in the UK and Canada. Out of 1809 histological type invasive ductal carcinoma (IDC) cases, 1453 samples for which microarray gene expression data were available are included in the analysis [33]. For these patient samples, ER, PR and HER2 expression status, gene expression data, overall survival, disease-specific survival and disease-free survival data were extracted from https://www.cbioportal.org/ (accessed on 19 December 2020). The IDC patients were further categorized into three subtypes: HR (hormone receptor)-positive (ER^+ve^, PR^+ve^, HER2^−ve^), HER2-positive (HER2^+ve^, ER^+ve/−ve^/PR^+ve/−ve^) and TNBC (ER^−ve^, PR^−ve^, HER2^−ve^) (Figure 1A).

TCGA (The Cancer Genome Atlas)–BRCA RNA-seq (FPKM-upper quartile normalized RNA-Seq) data (Illumina HiSeq 2000 RNA sequencing platform, 1218 samples) [34,35] and associated clinical data were downloaded from GDC using TCGAbiolinks package [36] in R 4.02 (R Core Team, 2020, Vienna, Austria) on 6 May 2021. Of 1218 breast cancer cases, 516 cases that were identified as IDC and IDC-like histological subtypes were included in the analysis. All gene expression data were log2 normalized before analysis. Out of 516, 451 cases were further categorized based on ER, PR and HER2 IHC receptor expression and HER2 FISH status into molecular subtypes. The three molecular subtypes are HR-positive (ER^+ve^, PR^+ve/−ve^, HER2^−ve^ or HER2 equivocal), HER2-positive (HER2^+ve^, ER^+ve/−ve^/PR^+ve/−ve^) and TNBC (ER^−ve^, PR^−ve^ and HER2^−ve^ or equivocal). The HER2-positive subtype includes ER/PR positive cases since the number of HER2^+ve^ and ER^−ve^ patients is limited to 23 in the TCGA dataset (Figure 1B).

### 2.2. YAP Target Gene Expression Correlation

YAP target gene signature derived from the mammary epithelial cell line MCF10a with stable overexpression of YAP was taken further for analysis [22]. The list of 62 YAP target genes is given in Appendix A. The expression of these YAP target genes was analyzed by Spearman correlation with YAP1 gene expression in both the IDC cohorts from METABRIC and TCGA. For METABRIC, microarray expression and TCGA–BRCA dataset with RNAseq were analyzed. The genes with Spearman correlation rho *p* ≥ 0.20 and *p*-value *p* < 0.05 were shortlisted for the IDC cohort as well as the three subtypes.

The correlation between all the YAP target genes was also calculated based on Pearson’s correlation coefficient using the normalized gene expression values using the R function ‘corr’ with significance at a *p*-value less than 0.05.

### 2.3. Correlation of YAP Target Genes and Clinical Parameters

Clinical parameters associated with each IDC patient, such as grade, stage, tumor size and lymph node status, were downloaded for both datasets. Patient samples were categorized into low (I–II) vs. high (III) grade (only available for METABRIC dataset), early (I–IIA) vs. late (IIB–IV) stage, small (pT1–T2) vs. large (pT3–T4) tumor size and negative vs. positive lymph node according to lymph node status. Gene expression for an individual gene was compared between the two categories. Welch two-sample test (unpaired *t*-test) was performed to look at significant variation in gene expression levels between the two categories for each clinical feature. Genes with a *p* < 0.05 and aggressive clinical features (high grade, late-stage, large size and positive lymph node) present with higher mean gene expression value for at least one of the clinical features were shortlisted for the IDC cohort as well as the three subtypes.

### 2.4. Cluster Analysis of Shortlisted YAP Signature Genes

For the shortlisted genes based on the Spearman correlation analysis and survival analysis, clustering was performed by hierarchical clustering. The R package ‘hclust’ was used, and the distance matrix was calculated by using the Euclidean method. All the rest of the parameters were set as standard. Clustering analysis was performed for HER2-positive and TNBC subtypes.

### 2.5. Survival Analysis

Overall survival (OS), disease-specific survival (DSS) and disease-free survival (DFS) analyses were performed. Overall survival (OS) was defined as the time from diagnosis until death. Disease-specific survival (DSS) was defined as the time from diagnosis until death, excluding patients who died from causes other than the disease. Disease-free survival (DFS) was defined as the time from the completion of primary treatment until clinical confirmation of tumor recurrence. The total follow-up period available for the METABRIC dataset is 355 months; for the TCGA dataset, it is 220 months, with follow-up of more than 120 months for only six patients. The survival analysis was performed using GraphPad Prism 5.

### 2.6. Survival Analysis for YAP1 Expression

Survival analysis for YAP1 based on median and quartile expression was performed for the IDC cohort and the three molecular subtypes. The Kaplan–Meier survival was plotted for eight years of follow-up with hazard ratio and significance.

### 2.7. Survival Analysis for YAP1 Target Gene Expression

Survival analysis was performed for high and low expression of each gene with median gene expression as the cutoff. Kaplan–Meier (KM) survival analysis was performed using the ‘survival’ package [37] in R program 4.02 for the shortlisted YAP1 target genes. Regression analysis was performed to define the hazard ratio. Shortlisting of genes was performed based on a hazard ratio >1.2 or <0.8 and a *p*-value of less than 0.05 for overall survival, disease-specific survival or disease-free survival.

### 2.8. Survival Analysis for YAP1 and YAP1 Target Gene Expression

A combined survival analysis of YAP1 and its shortlisted target gene was performed. Both the genes’ expression was divided into two subgroups groups, high and low, with median gene expression as a cutoff. The samples were divided into four groups as YAP1-high/target gene-high, YAP1-high/target gene-low, YAP1-low/target gene-high and YAP1-low/target gene-low.

### 2.9. Survival Analysis for YAP Target Gene Cluster

Based on the first division on the dendrogram of the hierarchical clustering analysis, the patients were divided into cluster 1 and cluster 2 for HER2-positive and TNBC subtypes. A Kaplan–Meier curve was plotted for patients in the two clusters. Overall survival and disease-free survival were performed for 60 months of follow-up as a cut-off.

### 2.10. Validation of Shortlisted Genes Using Independent Dataset

Two breast cancer GEO datasets were used for validating the genes shortlisted from the METABRIC and TCGA datasets. GEO datasets GSE20711 (n = 90) and GSE21653 (n = 213) with a total of 356 samples were considered as molecular subtype and long-term disease-free survival (average of 67 months) information was available. CEL files were downloaded using the GEOquery R package. Oligo R package was used to merge the datasets and normalize gene expression data. For genes with expression data using multiple probe sets, only those probes that recognize unique transcripts (ending with ‘_at’) were used and were further averaged if there were two or more probes recognizing unique transcripts for a particular gene [38]. Non-IDC and patients with no DFS and molecular subtype information (n = 66) were excluded. The validation cohort consisted of 291 samples, comprising 85 HR-positive, 98 HER2-positive and 89 TNBC samples. Survival analysis for YAP1 based on median and quartile expression was performed for the IDC cohort and the three molecular subtypes using Prism 5. The Kaplan–Meier survival was plotted for eight years of follow-up with hazard ratio and significance. Spearman correlation of YAP1 with the expression of shortlisted genes for HER2 and TNBC subtypes was performed. Survival analysis was performed for high and low expression for the shortlisted genes for HER2 and TNBC subtypes using the R program. A combined survival analysis of YAP1 and its shortlisted target genes was performed using Prism 5 for HER2 and TNBC subtypes. Hierarchical clustering was performed for the shortlisted genes for HER2 and TNBC subtypes using the R program, followed by survival analysis of the clusters obtained as mentioned above.

## 3. Results

YAP1 expression and its target signature of 22 genes are shown to be associated with poorer survival in solid tumors [11]. This comprehensive analysis of YAP and the Hippo pathway in various cancers was performed on a broader scale in all cancers, with the gene signature derived from the context of various cancer cell lines. To understand the association of YAP1 and its target expression in breast cancer, specifically in a subtype-specific manner, we attempted to analyze a YAP1 target gene signature derived from a mammary epithelial cell line [22] (Appendix A) for association with survival outcomes in breast cancer datasets. Clinical and gene expression data for patients with invasive ductal carcinoma (IDC), the most aggressive histological subtype, were downloaded for this analysis from two publicly available datasets; TCGA and METABRIC. The molecular subtype cohorts were identified based on annotated expression status of ER, PR and HER2 for each patient (Figure 1).

### 3.1. Survival Analysis for YAP1 Gene Expression

Survival analysis for the IDC cohorts and the three subtypes was performed for quartile (Figure 2) and median (Appendix A) expression cut-off of YAP1. For the IDC cohort, the METABRIC dataset showed a significant association of YAP1 expression with better outcomes compared to TCGA, where YAP1 expression was associated with poorer survival outcomes, though not significantly (Figure 2A). Subtype-wise analysis of YAP1 expression showed a concordant association for HR-positive and HER2-positive subtypes for both datasets. YAP1 expression is associated with better outcomes in HR-positive subtypes (Figure 2B), while it is associated with worse outcomes in the HER2-positive subtype (Figure 2C). For the TNBC subtype, the METABRIC dataset did not show any specific association with YAP1 expression, while subtype cohorts from TCGA showed a trend towards poorer outcomes (Figure 2D). Since METABRIC and TCGA datasets revealed different patterns of association with YAP1 expression for survival outcomes, the two datasets were further analyzed as independent cohorts for YAP1 target gene signature.

### 3.2. YAP1 and YAP1 Target Gene Expression Correlation

YAP1 protein is translocated between the cytoplasm and the nucleus under the regulation of the Hippo signaling pathway [10]. Nuclear translocation of YAP1 leads to activation of the YAP1 target gene signature that promotes oncogenic phenotypes [22].To account for the transcriptionally active form of YAP1, the YAP1 target gene signature derived from a mammary epithelial cell line was analyzed within the IDC and subtype cohorts with respect to YAP1 expression.

Spearman correlation was performed for each target gene with respect to YAP1 expression for the two IDC cohorts and the three subtypes for each dataset. The correlation coefficient (rho) values and *p*-values for all the genes with respect to YAP1 expression are listed in Appendix A. Genes with a Spearman correlation coefficient (rho) of more than 0.20 and a significance of less than 0.05 were shortlisted for each cohort. The numbers of genes shortlisted for the IDC cohort and the three subtypes within METABRIC and TCGA datasets are given in Table 1. Correlation plots of representative shortlisted genes are shown in Figure 3 for METABRIC and TCGA datasets according to the subtypes.

Correlation between all the YAP1 target genes with respect to each other was computed for both datasets. The correlation matrix for Pearson coefficients is plotted for the signature for the IDC cohorts (Appendix A), HR-positive subtype (Appendix A), HER2-positive subtype (Figure 4A) and TNBC subtype (Figure 4B). Each of the subtypes in both datasets revealed differential association, with the strong concordance for ECM-related genes (highlighted in peach) within the HER2-positive and TNBC subtypes (Figure 4).

### 3.3. YAP1 Target Genes and Association with Clinical Parameters

The shortlisted target genes that showed significant concordance with YAP1 expression were further analyzed for association with aggressive clinical parameters such as high grade, larger tumor size, later stage and lymph node positivity. Patient samples were categorized into low (I–II) vs. high (III) grade (only available for METABRIC dataset), early (I–IIA) vs. late (IIB–IV) stage, small (pT1–T2) vs. large (pT3–T4) tumor size and negative vs. positive lymph node according to lymph node status. Gene expression levels between the two groups for each clinical feature were compared for mean difference with Student’s *t*-test. Genes with a *p*-value less than 0.05 and aggressive clinical features present with a higher mean gene expression value for at least one of the clinical features were shortlisted. The *p*-values and differences in mean gene expression for each clinical parameter are listed in Appendix A for METABRIC and TCGA datasets, respectively. The numbers of genes shortlisted for the METABRIC dataset are given in Table 2. No genes showed significant association with the aggressive clinical features for the TCGA dataset.

### 3.4. Survival Analysis

The shortlisted target genes that showed significant concordance with YAP1 expression and significant association with the aggressive clinical features were further analyzed for association with survival outcomes. Kaplan–Meier survival curves were plotted for each gene based on median expression cut-off for a 60-month follow-up period for the shortlisted genes for overall survival (OS), disease-specific survival (DSS) and disease-free survival (DFS). Cut-off of median gene expression was considered instead of quartile gene expression for a stringent shortlisting of the YAP1 target genes. The shortlisted genes’ hazard ratio and *p*-values for survival analysis are tabulated in Appendix A. Further shortlisting was performed based on the hazard ratio of more than 1.2 or less than 0.8 and a *p*-value of less than 0.05 for any one of the three survival analyses. Shortlisting was performed for IDC and the three molecular subtype cohorts. The final list of shortlisted genes is tabulated in Table 3. Representative survival graphs for METABRIC and TCGA databases for IDC and the three subtypes are shown in Figure 5. High expression of the shortlisted genes was associated with aggressive clinical features only for the METABRIC dataset. High expression levels of ADAMTS1, FN1 and IGFBP3 are associated with positive lymph node status. In addition, poor survival outcomes were observed for patients with high expression of these three genes for the IDC cohort of the METABRIC dataset (Appendix A).

For the shortlisted genes, survival analysis was performed to assess the association with survival outcomes when their expression was considered along with YAP1 expression. Overall survival and disease-free survival analyses were performed for a 60-month follow-up period for HER2-positive and TNBC subtypes, as the most significant association was observed for these subtypes for the individual genes. Within the METABRIC dataset, COL12A1 and PROS1 were shortlisted for the HER2-positive subtype (Figure 6A) and COL12A1 and SULF1 were shortlisted for the TNBC subtype (Figure 6B). For both the genes, for both the subtypes, low expression with high YAP1 expression showed the worst outcome trends (Figure 6). Similarly, for the TCGA dataset, the three genes shortlisted for HER2-positive subtype are COL12A1, FN1 and SULF1. Unlike METABRIC, for the TCGA dataset, high COL12A1 expression with low YAP1 expression was associated with the worst outcomes, while low COL12A1 expression and low YAP1 expression were associated with better survival (Figure 7A). FN1 and SULF1 showed an association with survival outcomes similar to that of COL12A1 (Figure 7A). For the TNBC subtype in the TCGA dataset, THBS1 was the only gene that was shortlisted. A subset with high-YAP1/high-THBS1 expression was associated with significantly worse overall survival, while low-YAP1/low-THBS1 had good survival outcomes (Figure 7B).

### 3.5. Hierarchical Clustering of Shortlisted Genes for Survival Analysis

Hierarchical clustering was performed for the shortlisted genes listed in Table 3 for HER2-positive and TNBC subtypes. Based on the dendrogram, the cohorts were grouped into two expression clusters. For the HER2-positive subtype, in the METABRIC dataset, clustering was performed for COL12A1 and PROS1 expression (Figure 8A). Cluster 1, where the expression level of COL12A1 is high while that of PROS1 is low, was associated with poorer survival than cluster 2. For TNBC, subtype cluster analysis was performed for the shortlisted genes SULF1 and COL12A1 (Figure 8B). High SULF1 and COL12A1 expression aligned with cluster 1, which is associated with poorer outcomes than cluster 2 with low expression of the two genes.

Within the TCGA database, three genes were shortlisted for the HER2-positive subtype, namely FN1, COL12A1 and SULF1 (Figure 8C), and only one gene was shortlisted for the TNBC subtype. For the HER2 subtype, the two clusters were segregated with relatively high expression of the three genes in cluster 1 compared to cluster 2. Cluster 1, with high expression of the genes, was associated with poorer disease-free survival outcomes (Figure 8C).

As observed here, the shortlisted genes, similar to YAP1, show opposing association trends with survival outcomes in METABRIC and TCGA datasets.

### 3.6. Validation of the Shortlisted Genes in an Independent Cohort

Two GEO breast cancer datasets (GSE21653 and GSE20711) with 291 samples were merged to validate the prognostic value of the shortlisted genes for HER2-positive (*n* = 98) and TNBC (*n* = 89) subtype. These datasets were considered as they included molecular subtype and long-term DFS follow-up information. Survival analysis (DFS) for YAP1 was performed for IDC and the three subtypes to see if the outcomes in the independent cohort are similar to those in METABRIC and TCGA. As observed in Figure 9, high YAP1 expression was associated with poor DFS for HER2-positive and TNBC subtypes, while better outcomes were seen for the HR-positive subtype. Similar survival outcomes were observed for the TCGA dataset (Figure 2).

To validate the prognostic potential of the shortlisted YAP1 target genes (Table 3) Spearman correlation and survival analysis was performed for HER2-positive and TNBC subtypes using the independent dataset (Figure 10). As mentioned in Table 3, four genes were shortlisted for the HER2-positive subtype (COL12A1, PROS1, SULF1, FN1), and three genes were shortlisted for the TNBC subtype (COL12A1, SULF1, THBS1). Significant correlation with a rho value of more than 0.2 was observed for all four shortlisted genes within the HER2 subtype (Figure 10A), as observed in METABRIC and TCGA datasets. For the TNBC subtype, the rho values for SULF1 and THBS1 were 0.2, but COL12A1 with a rho value of 0.18 did not pass the cut-off (Figure 10B). A poor correlation between YAP1 and shortlisted YAP1 target gene expression was observed in the independent cohort than in the METABRIC and TCGA cohort within the TNBC subtype.

Disease-free survival curves were plotted for the shortlisted genes based on median expression cut-off for a 60-month follow-up period using the independent cohort for HER2-positive and TNBC subtypes. High COL12A1 expression was associated with significantly poor disease-free survival outcomes within the HER2-positive subtype in the independent cohort, as observed in the TCGA dataset. Survival outcomes of PROS1 show the same trend as the HER2-positive METABRIC cohort, and SULF1 and FN1 follow the same trend as TCGA, although they are insignificant (Figure 11). When accessing the survival outcomes of the shortlisted YAP1 targets along with YAP1, high-YAP1/high-target-gene expression was associated with worse survival outcomes for all four shortlisted genes within the HER2 subtype (Figure 11). These findings suggest that high COL12A1, PROS1, SULF1 and FN1 expression levels are associated with poor survival outcomes for HER2-positive patients and exhibit worse outcomes in presence of high YAP1 expression. As shown in Figure 7A, high YAP1 and high COL12A1/SULF1/FN1 expression levels are associated with worse survival in the HER2-positive TCGA cohort, which was also observed in the independent cohort. High-YAP1/high-PROS1 expression is also associated with poor survival in the HER2-positive METABRIC cohort (Figure 6A). The low-YAP1/low-COL12A1 expression group exhibits worse outcomes in the HER2-positive METABRIC cohort (Figure 6A), which is opposite to what we see in the independent cohort (Figure 11).

In the case of the TNBC subtype, high expression levels of COL12A1, SULF1 and THBS1 were associated with poor DFS but were insignificant (Figure 12). THBS1 follows the same trend as observed in the TCGA cohort, but both COL12A1 and SULF1 show opposite survival outcomes to that observed in the METABRIC dataset within the TNBC subtype. Furthermore, patients with high COL12A1 along with high YAP1 expression exhibit worse outcomes in the independent cohort (Figure 12), which does not overlap with our findings within the METABRIC dataset (Figure 6B). No clear separation was observed in the case of DFS for SULF1 and THBS1 along with YAP1 within the TNBC subtype for the independent cohort (Figure 12).

Hierarchical clustering was performed for the shortlisted genes listed in Table 3 for HER2-positive and TNBC subtypes using the independent cohort. Based on the dendrogram, the cohorts were grouped into two expression clusters. As differential expression was observed in cluster 1, it was further divided into two or three clusters (Figure 13). As for the HER2-positive subtype, clustering was performed for genes shortlisted based on METABRIC and TCGA separately. For clustering performed for COL12A1 and PROS1 expression (Figure 13A) within the HER2-positive subtype, cluster 1, where the expression level of COL12A1 is high and PROS1 is high/low, was associated with poorer survival than cluster 2. When cluster 1 was further divided into three groups, it was observed that patients with high COL12A1 and PROS1 are associated with worse survival. These findings do not overlap with the HER2-positive METABRIC cohort cluster of COL12A1 and PROS1 (Figure 8A), where low COL12A1 expression and high PROS1 expression are associated with poor survival. For clustering performed for COL12A1, SULF1 and FN1 expression (Figure 13B) for the HER2 subtype, cluster 2 of the four clusters, where the expression levels of COL12A1, SULF1 and FN1 are high, was associated with worse survival. Similar outcomes were observed for the HER2-positive TCGA cohort (Figure 8C), indicating the survival pattern for the independent cohort and TCGA cohort are alike in the case of the HER2-positive subtype. For TNBC, subtype cluster analysis was performed for the shortlisted genes, SULF1 and COL12A1 (Figure 13C). High SULF1 and COL12A1 expression aligned with cluster 1, which is associated with better outcomes than cluster 2 with low expression of the two genes. The outcomes obtained for the TNBC subtype are opposite to what was observed in the METABRIC dataset (Figure 8B).

## 4. Discussion

Multiple gene signatures have been identified for breast cancer to predict aggressive and treatment-responsive cancers. Gene signatures such as MammaPrint (70-gene signature), 21-gene signature (Oncotype DX assay) and PAM50 are used for prognostic and diagnostic purposes and are FDA-approved for better patient management. The 21-gene signature is used to predict the possibility of distant recurrence in node-negative ER-positive breast cancer patients priorly treated with tamoxifen [39]. Oncotype DX is a 16-gene signature that estimates the likelihood of recurrence in ER-positive breast cancer patients and is used to determine the necessity of chemotherapy administration. Low scores of Oncotype DX can guide the least likelihood of recurrence, and such patients can be spared from unnecessary chemotherapy [40]. PAM50 is a 50-gene signature used for classifying breast cancer into five intrinsic subtypes—luminal A, luminal B, HER2 enriched, basal and normal-like [41]. MammaPrint assay is a microarray-based multigene (70 genes) assay for ER^+/−^ node-negative breast cancer patients to determine the likelihood of recurrence and based on which treatment administration is determined [42].

MammaPrint, a 70-gene signature, divides patients into two groups, high risk and low risk for recurrence. The low-risk group is further divided into low-risk and ultra-low-risk groups. In phase 3 of the MINDACT trial, 6693 patients participated, of which 1000 were classified under the ultra-low-risk group. The eight-year disease-specific survival for these patients was more than 99%. In comparison, the eight-year distant-metastasis-free survival was more than 97%, indicating an excellent prognosis for patients in the ultra-low-risk group [43]. The use of Oncotype DX has led to decreased administration of adjuvant chemotherapy and an increase in survival and up to 40% lower risk of breast-cancer-specific death [44]. The impact of these gene signatures test has helped clinical management of ER-positive breast cancer with a personalized approach. To our knowledge, there are limited studies focusing on identifying YAP1 target gene signatures such as for oral carcinoma and Ewing sarcoma [45,46]. Furthermore, there are no studies concentrating on breast cancer with respect to the YAP target signature. As mentioned above, commercially available gene signatures such as Oncotype DX are specific for ER-positive breast cancer, while the PAM50 signature is useful for identifying intrinsic breast cancer subtypes. Therefore, our study focusing on the YAP1 target gene signature specifically addressing HER2-positive and TNBC subtypes will be beneficial in advancing personalized therapy and identifying YAP1 target genes with potential prognostic value.

In this study, YAP1 target gene signatures were further analyzed to shortlist genes dependent on YAP1 expression in tumor samples that are significantly associated with recurrent breast cancers. The YAP1 target gene signature derived from stable mammary epithelial cell lines was thus further refined based on tumor tissue gene expression correlation with YAP1 and association with aggressive clinical features and survival outcomes of breast cancer patients in two publicly available datasets; METBRIC and TCGA. The survival outcomes for YAP1 expression and many of the target genes showed opposing trends in TCGA and METABRIC datasets. The two different platforms for gene expression assessment and patient demographics from different countries between the two datasets could contribute to the differential association observed [47]. Due to this differential association, the shortlisting of YAP1 target gene expression with the clinical features and survival outcomes was performed independently for both datasets. Stringent screening for correlation of expression with YAP1 and significant association with the clinical features and survival outcomes was performed to shortlist YAP1 target genes.

Out of the 62 YAP1 target genes that were identified from mammary epithelial cell lines, a few genes stood out as significantly associated with YAP1 expression, aggressive clinical features and worse survival outcomes. YAP has been shown to play an oncogenic role in solid cancers, including breast cancer. A few studies have reported YAP1 as a tumor suppressor in breast cancer, with high YAP expression associated with better survival outcomes [48,49,50]. Further characterization of the association of YAP1 with breast cancer according to ER expression has revealed that the role of YAP1 depends on its nuclear expression and molecular subtype, where high nuclear YAP1 expression is associated with aggressive clinical features in poor survival outcomes in the ER-negative subtype [51,52,53]. Furthermore, high YAP1 expression is significantly associated with poor distant-metastasis-free survival in TNBC [51]. Lehn et al., in their study, analyzed how YAP1 behaves differently in ER-positive and ER-negative breast cancer. Association of YAP1 with clinical and pathological data showed low YAP1 expression was associated with poor survival in ER-positive breast cancer. Further, it was also shown that YAP1 may be an important factor for sensitivity to endocrine therapies where the absence of YAP1 is associated with impaired tamoxifen response. In vitro studies using luminal cell lines showed that knockdown of YAP1 resulted in decreased sensitivity to tamoxifen, indicating the role of YAP1 in predicting outcomes in this ER-positive subgroup [53]. A similar trend was observed in the METABRIC and TCGA datasets as well as the independent dataset in our study where high YAP1 expression was associated with better survival outcomes in HR-negative breast cancer. Since YAP1 itself showed a strong trend towards worse survival outcomes, specifically in HER2-positive and TNBC subtypes, clustering analysis for shortlisted target genes was performed for these two subtypes to identify a progressive disease subset. Two genes were shortlisted from the METABRIC dataset for the HER2-positive (COL12A1 and PROS1) and TNBC (COL12A1 and SULF1) subtypes. From the TCGA dataset, three genes (COL12A1, SULF1, FN1) were shortlisted for the HER2-positive subtype, while only one gene (THBS1) survived the stringent selection criteria for the TNBC subset. Overall, COL12A1 is the only gene common through both datasets and subtypes.

Shortlisted genes when validated in the independent dataset showed similar results to those seen for the TCGA dataset, especially for the HER2 subtype. High COL12A1 expression was associated with poor survival in the independent cohort. YAP1 target COL12A1 (collagen type 12 alpha one chain) is an extracellular matrix protein [54] and is shown to be associated with progressive and recurrent cancers in multiple solid tumors. The extent of ECM component expression is determined by the percent tumor and stroma composition in the tumor tissue at the time of transcriptome sequencing and the depth of the sequencing. These could be a few confounding factors that led to the opposing association with the survival outcomes when compared with METABRIC. Nevertheless, an experimental investigation will help understand the direction of the association of COL12A1 with tumor progression.

In a study by Verghese et al. (2013), it was observed that high COL12A1 expression was associated with significantly increased recurrence in breast cancer patients [31]. Overexpression of COL12A1 was significantly correlated with poor prognosis in pancreatic adenocarcinoma patients. Furthermore, analysis of these patients shows the involvement of COL12A1 in epithelial–mesenchymal transition, focal adhesion formation and strong correlation of immune cell infiltration [55]. High COL12A1 expression is associated with poor overall survival and progression-free survival in patients with gastric cancer. In addition, aggressive clinical features such as advanced tumor stage, positive lymph node status and distant metastasis were observed in patients with COL12A1 overexpression [30].

Like COL12A1, other shortlisted candidate genes along with YAP1 need to be validated for their significant association with survival outcomes of HER2-positive and TNBC subtypes. The expression of the shortlisted genes can also be analyzed with respect to response to treatment towards pathological complete response, survival outcomes and clinical parameters in breast cancer patient samples in the Indian cohort. Successful validation in a larger cohort for breast cancer samples can take these shortlisted genes forward in development into a prognostic signature to predict treatment response and survival outcomes.

## Figures and Tables

**Figure 1 jcm-11-01947-f001:**
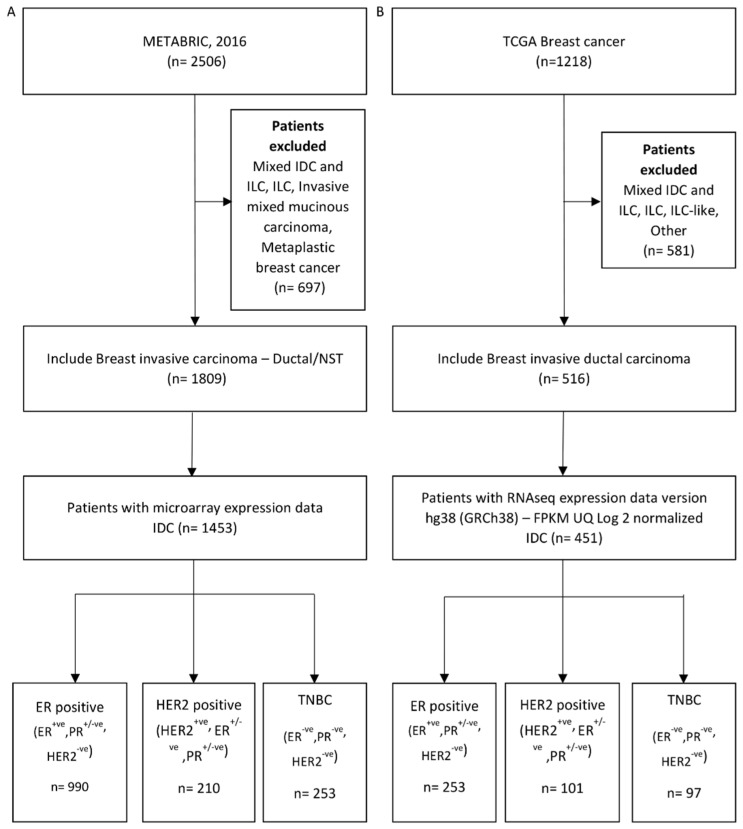
Patient cohort selection from METABRIC and TCGA. Flow chart depicting IDC and molecular subtype cohort selection process from METABRIC (**A**) and TCGA (**B**) datasets.

**Figure 2 jcm-11-01947-f002:**
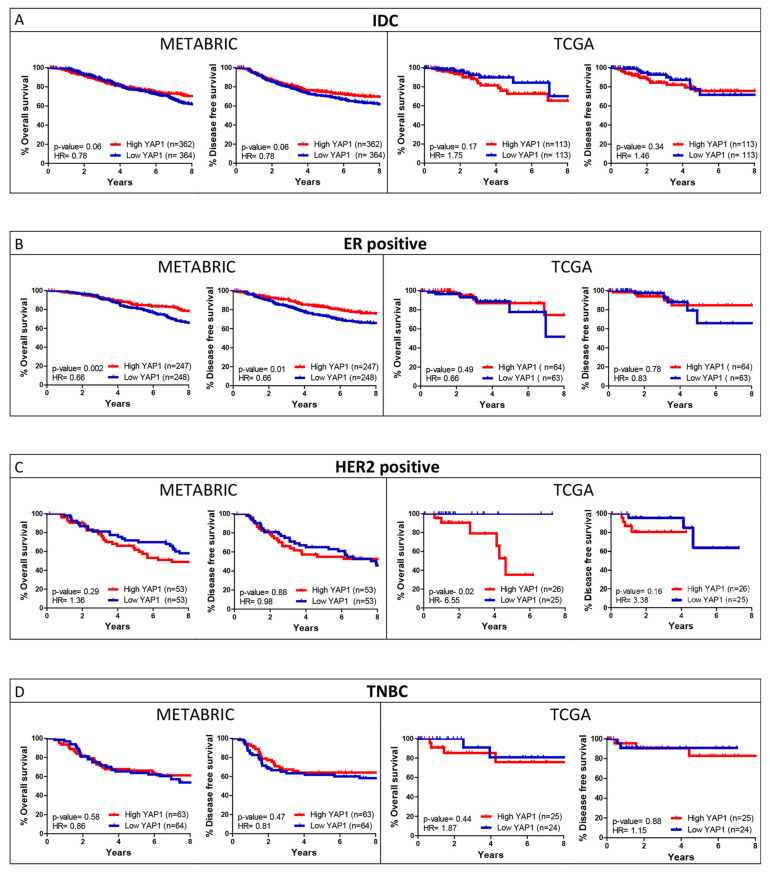
YAP1 expression and its association with OS and DFS outcomes of breast cancer patients. Kaplan–Meier survival curves are plotted for overall survival and disease-free survival over eight years of follow-up for YAP1 expression (Quartile) for IDC (**A**) and subtype (**B**–**D**) cohorts from METABRIC and TCGA datasets.

**Figure 3 jcm-11-01947-f003:**
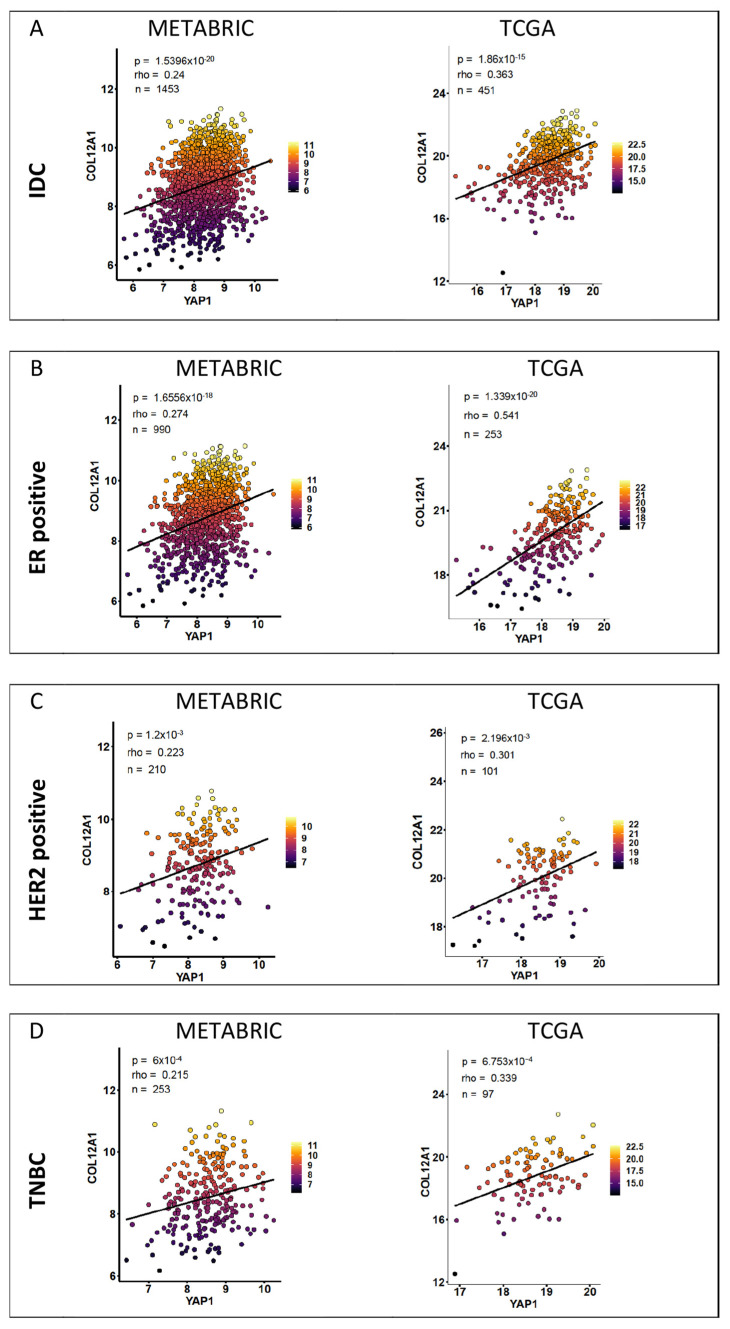
Correlation of YAP1 target gene and YAP1 expression within breast cancer patient samples. Representative correlation plots for significant Spearman correlation for YAP1 target gene expression with respect to YAP1 expression in tumors from IDC (**A**) and subtype cohorts (**B**–**D**) of METABRIC and TCGA datasets.

**Figure 4 jcm-11-01947-f004:**
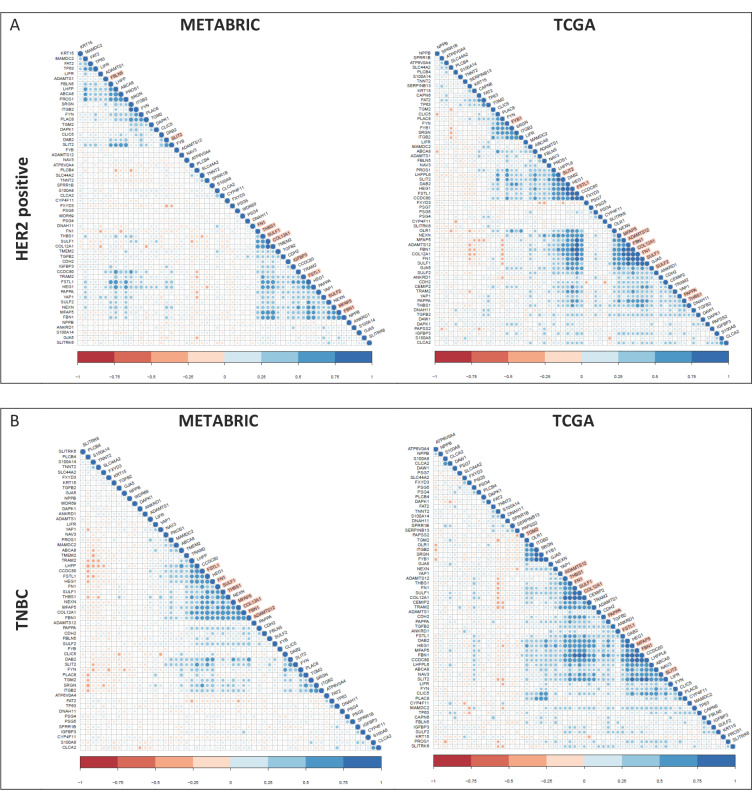
Correlation of YAP1 target genes with each other for their expression in subtype cohorts. Gene expression of YAP1 targets correlated with each other for their expression in breast cancer tumors analyzed within the HER2-positive (**A**) and TNBC (**B**) subtypes.

**Figure 5 jcm-11-01947-f005:**
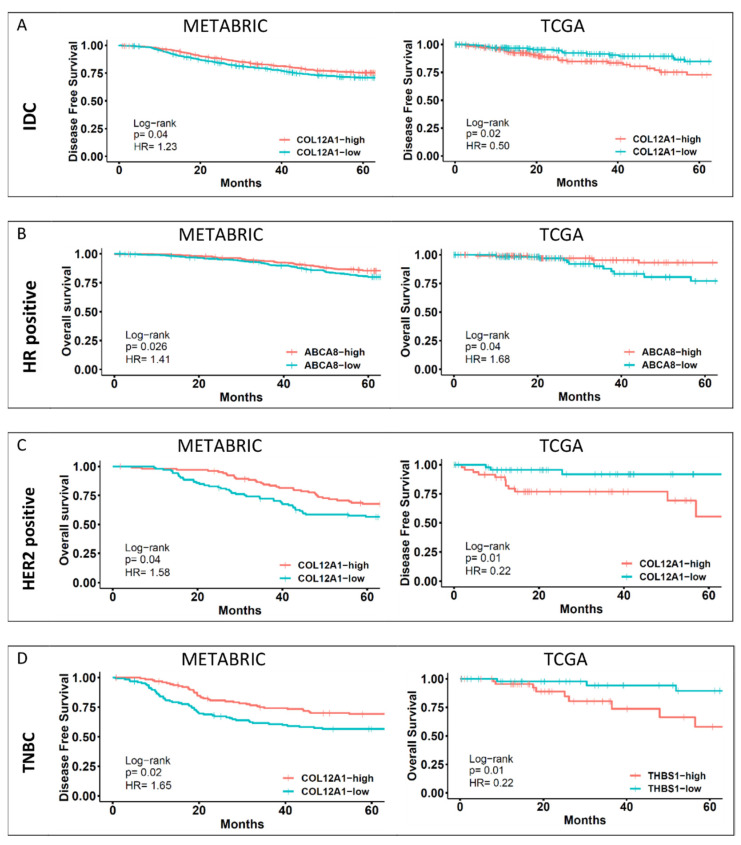
Subtype-wise survival outcomes of representative YAP1 target genes. YAP1 target genes were analyzed for their significant association with survival outcomes of IDC and subtype cohorts. Disease-free survival plots for the representative genes from the shortlist for each cohort are shown here. Association of COL12A1 expression in IDC (**A**), ABCA8 expression in HR-positive subtype (**B**) and COL12A1 expression in HER2-positive subtype (**C**) for both datasets—METABRIC and TCGA. Within the TNBC cohort, COL12A1 and THBS1 were considered for METABRIC and TCGA datasets (**D**), respectively.

**Figure 6 jcm-11-01947-f006:**
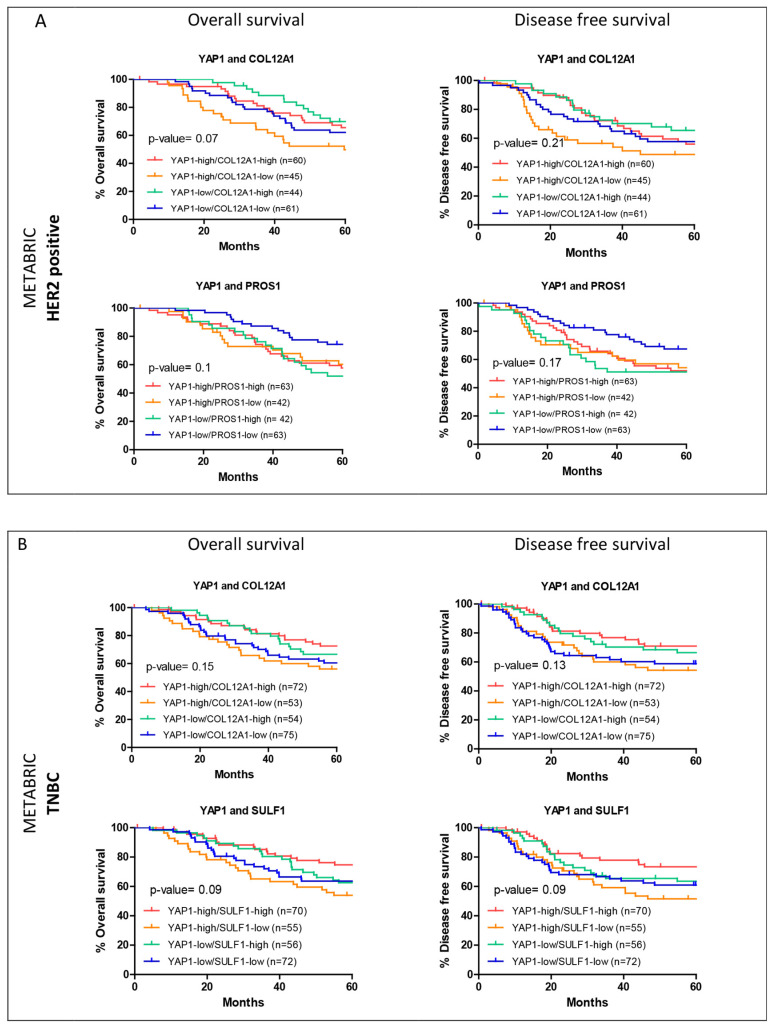
Survival curves for the expression of two genes within subtype cohorts from METABRIC. Shortlisted YAP1 target gene expression and YAP1 expression were analyzed for their association with overall and disease-free survival outcomes within their respective cohorts, HER2-positive (**A**) and TNBC (**B**) from METABRIC.

**Figure 7 jcm-11-01947-f007:**
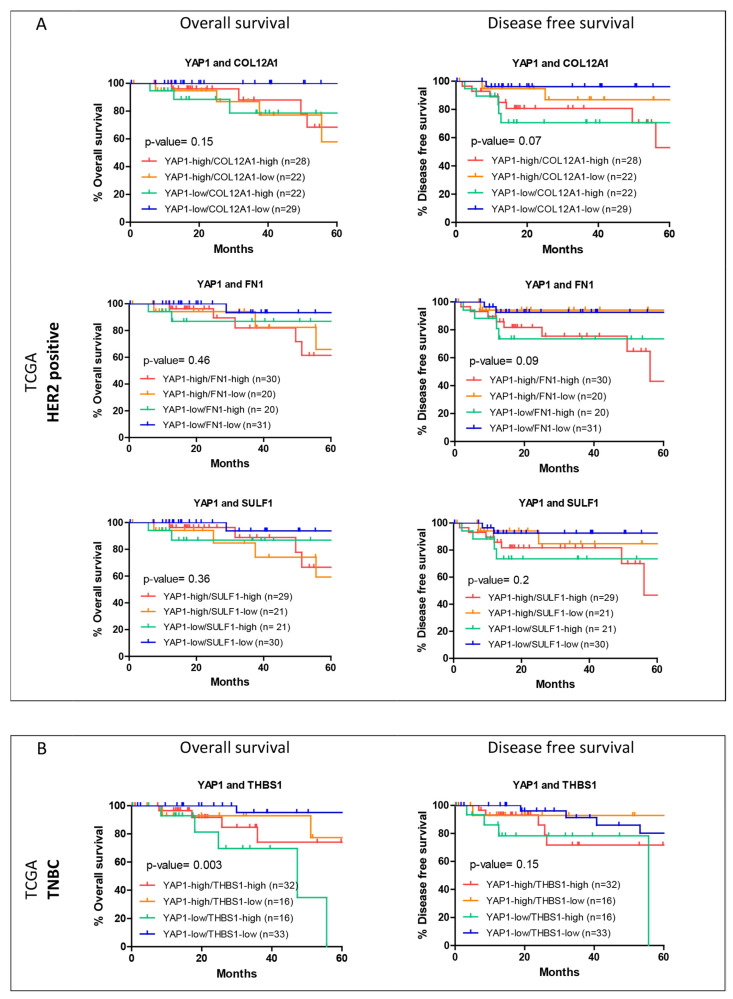
Survival curves for the expression of two genes within subtype cohorts from TCGA. Shortlisted YAP1 target gene expression and YAP1 expression were analyzed for their association with overall and disease-free survival outcomes within their respective cohorts, HER2-positive (**A**) and TNBC (**B**) from TCGA.

**Figure 8 jcm-11-01947-f008:**
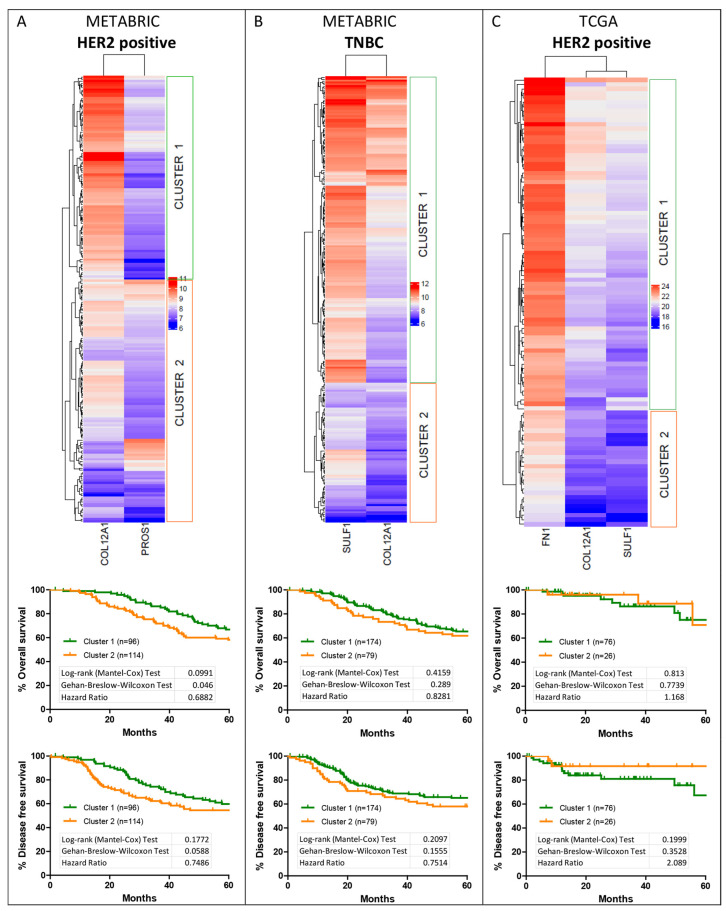
Hierarchical clustering for the shortlisted genes. Hierarchical clustering was performed for the shortlisted genes for their expression within the respective subtype cohort, HER2-positive from METABRIC dataset (**A**) and TCGA dataset (**C**) and TNBC from METABRIC dataset (**B**). The primary two clusters were analyzed for their association with the overall and disease-free survival within the subtype cohorts.

**Figure 9 jcm-11-01947-f009:**
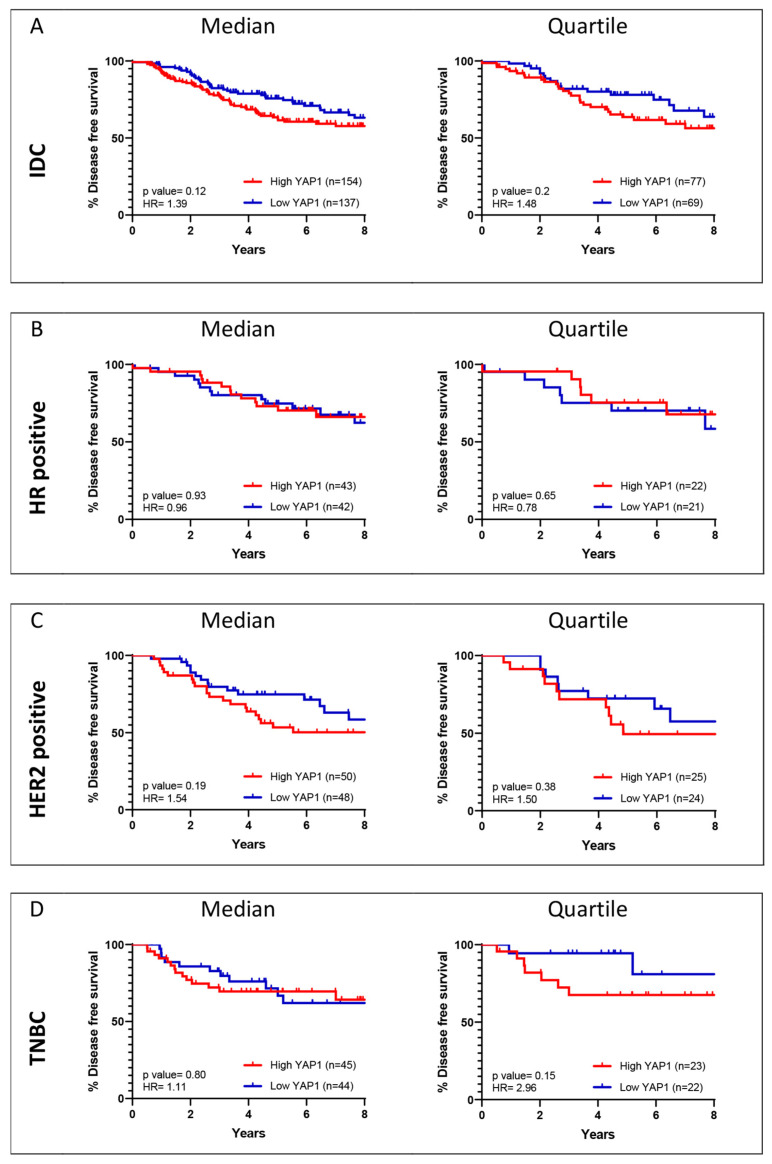
YAP1 expression and its association with DFS outcomes of an independent breast cancer cohort. Kaplan–Meier survival curves are plotted for disease-free survival over eight years of follow-up for YAP1 expression (median and quartile) for IDC (**A**) and subtype (**B**–**D**) cohorts.

**Figure 10 jcm-11-01947-f010:**
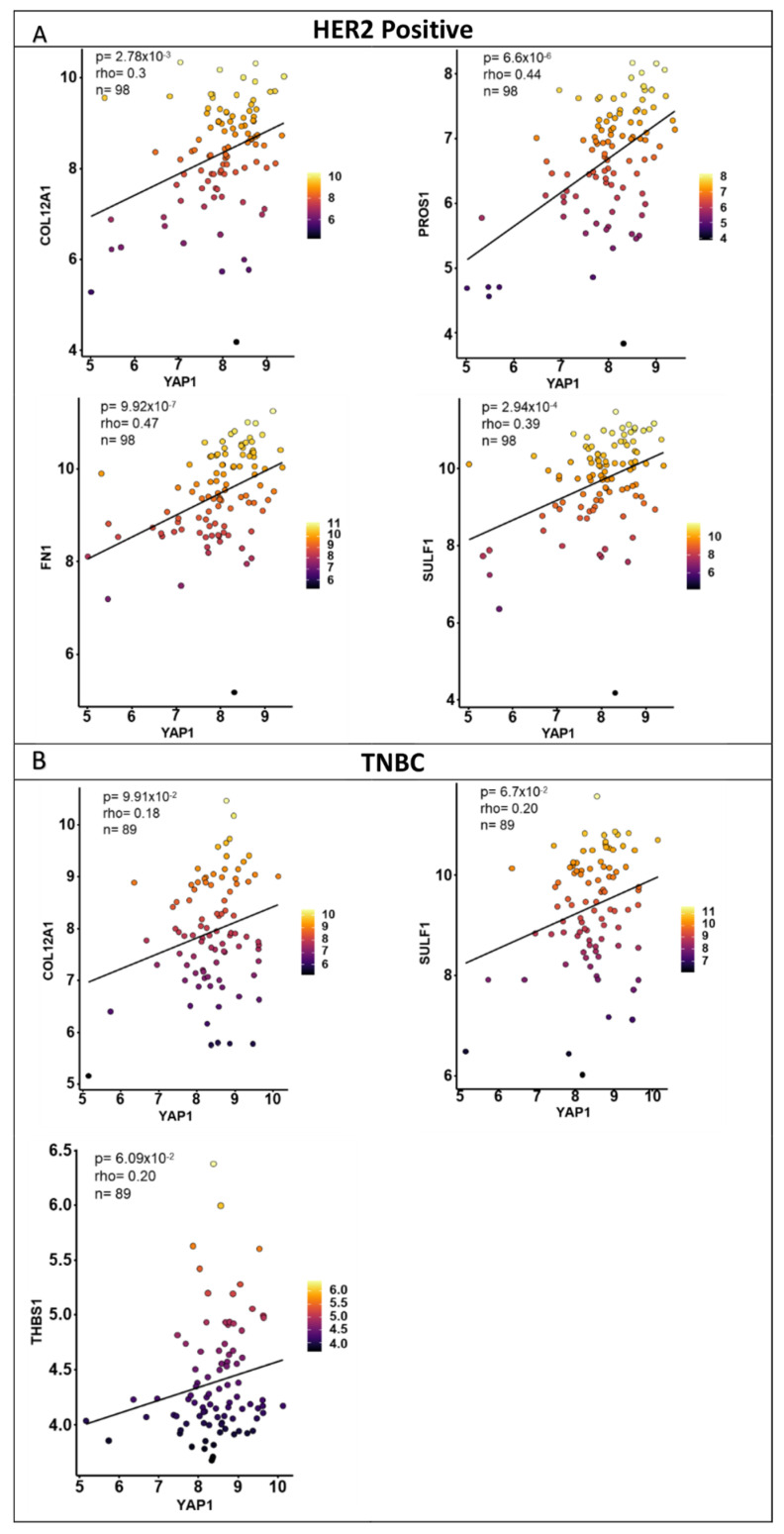
Correlation of shortlisted YAP1 target gene and YAP1 expression within the independent breast cancer patient samples. Correlation plots with significant Spearman correlation for YAP1 target gene expression with respect to YAP1 expression for COL12A1, PROS1, FN1 and SULF1 within the HER2 subtype and for COL12A1, SULF1 and THBS1 within the TNBC subtype.

**Figure 11 jcm-11-01947-f011:**
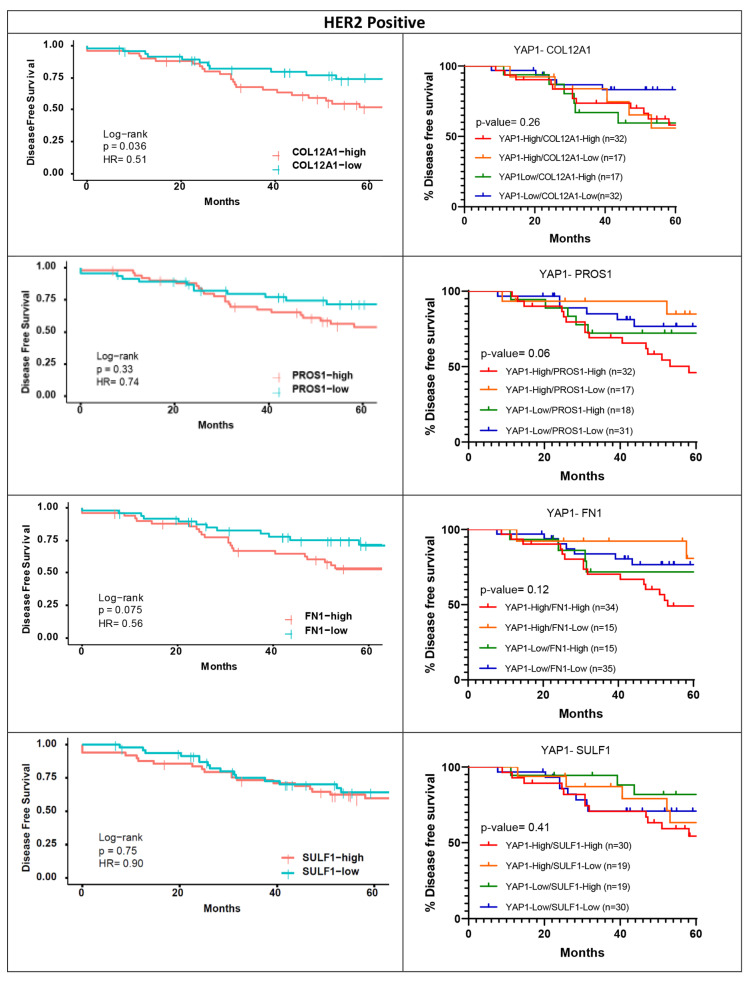
DFS outcomes and two gene survival outcomes for the shortlisted YAP1 target genes with HER2-positive subtype for the independent cohort. DFS plots for COL12A1, PROS1, FN1 and SULF1 (**left**). Shortlisted YAP1 target gene expression and YAP1 expression were analyzed for their association with disease-free survival outcomes for the HER2-positive subtype.

**Figure 12 jcm-11-01947-f012:**
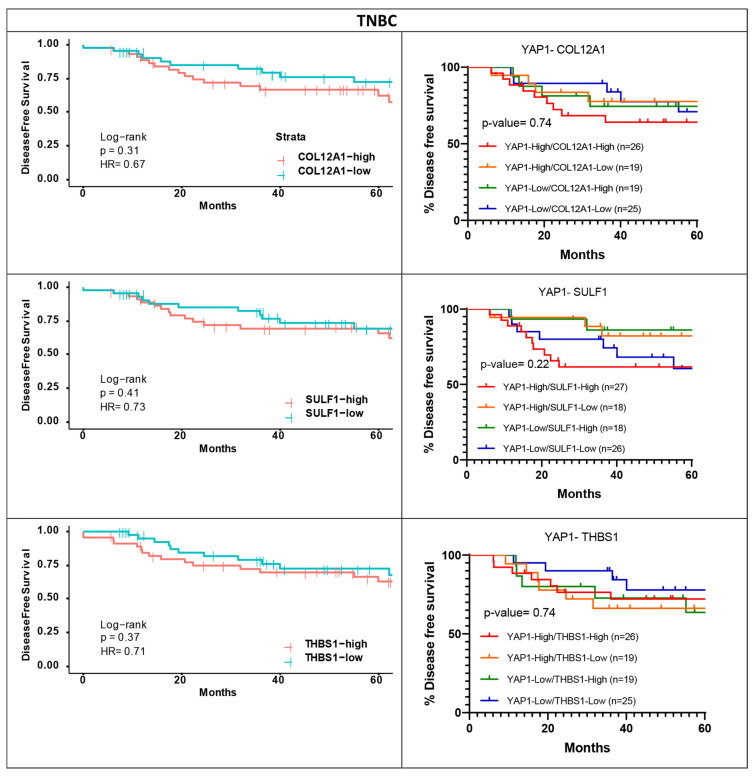
DFS outcomes and two gene survival outcomes for the shortlisted YAP1 target genes with TNBC subtype for the independent cohort. DFS plots for COL12A1, SULF1 and THBS1 (**left**). Shortlisted YAP1 target gene expression and YAP1 expression were analyzed for their association with disease-free survival outcomes for the TNBC subtype.

**Figure 13 jcm-11-01947-f013:**
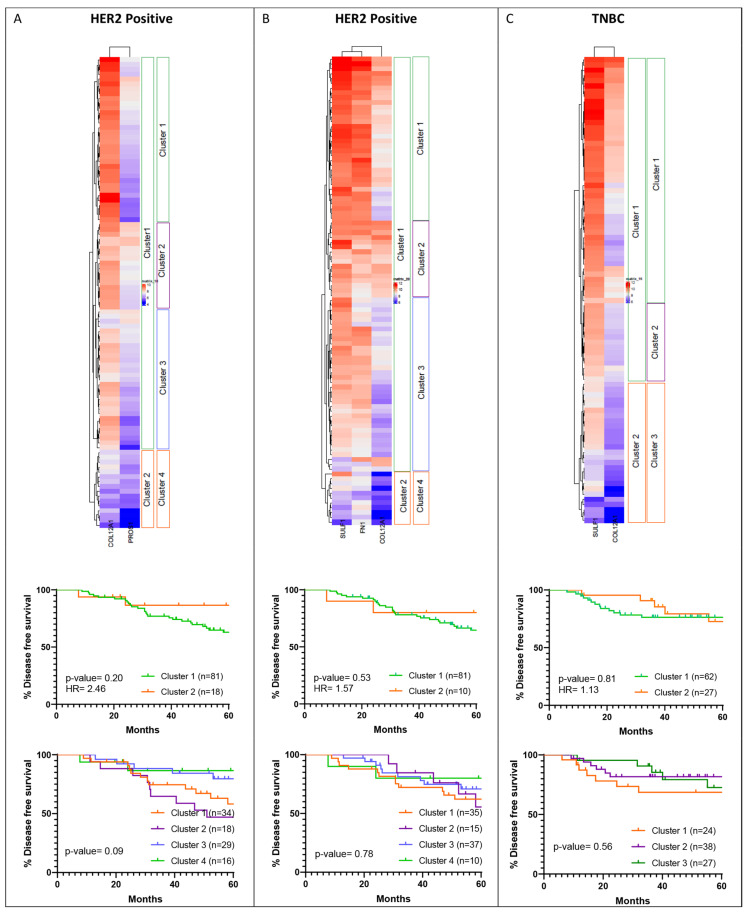
Hierarchical clustering for the shortlisted genes. Hierarchical clustering was performed for the shortlisted genes from the METABRIC HER2 cohort (**A**), TCGA HER2 cohort (**B**) and METABRIC TNBC cohort (**C**) using the independent cohort for HER2 and TNBC subtypes. The highlighted clusters were analyzed for their association with disease-free survival within the subtype for the independent cohort.

**Table 1 jcm-11-01947-t001:** Total numbers of genes shortlisted based on expression correlation with YAP1 expression in breast cancer samples, analyzed according to subtypes within two datasets.

	METABRIC	TCGA
IDC	23	27
HR-Positive	26	34
HER2-Positive	24	21
TNBC	13	18

**Table 2 jcm-11-01947-t002:** Total numbers of genes shortlisted based on the association of their expression with the clinical parameters available within the METABRIC dataset.

	METABRIC
IDC	9
HR-Positive	5
HER2-Positive	3
TNBC	1

**Table 3 jcm-11-01947-t003:** A list of shortlisted genes, where their expression is significantly associated with survival outcomes of breast cancer patients of IDC and subtype cohorts from METABRIC and TCGA datasets.

Genes Shortlisted Based on Survival Analysis							
METABRIC				TCGA			
IDC (*n* = 7)	HR-Positive (*n* = 5)	HER2-Positive (*n* = 2)	TNBC (*n* = 2)	IDC (*n* = 8)	HR-Positive (*n* = 10)	HER2-Positive (*n* = 3)	TNBC (*n* = 1)
ABCA8	ABCA8	COL12A1	COL12A1	CAPN6	ABCA8	COL12A1	THBS1
ADAMTS1	FBLN5	PROS1	SULF1	COL12A1	CAPN6	FN1	
COL12A1	MAMDC2			DAPK1	COL12A1	SULF1	
FN1	SULF1			FBN1	DAB2		
IGFBP3	THBS1			FN1	FBN1		
LIFR				FSTL1	FN1		
MAMDC2				MFAP5	LIFR		
				THBS1	MFAP5		
					PROS1		
					SULF1		

## Data Availability

Code files are available at https://github.com/doc-dk/gomathi-etal (accessed on 22 January 2022).

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
