# Peer review of "Analysis of Yes-Associated Protein-1 (YAP1) Target Gene Signature to Predict Progressive Breast Cancer"

_jcm, 2022, doi:10.3390/jcm11071947_

Round 1

Reviewer 1 Report

High YAP1 expression and its transcriptional targets are associated with poor survival for many types of cancer (PMID: 30380420), and its functional relevance has been demonstrated in many studies, including in breast cancer. In this study, Gomathi et al investigate expression of YAP1 target genes in two public databases (METABRIC and TCGA) and attempt to link YAP1 target gene signatures with the outcome of breast cancer patients broken down into molecular subtypes. A very limited number of these target genes showed significant association with survival outcomes of HER2 23 positive and TNBC patients in both the datasets.

Enthusiasm for this study is very limited. Most random gene expression signatures are significantly associated with breast cancer outcome (PMID: 22028643). Reasons for this include the fact that (1) the expression hundreds of genes is correlated with survival; (2) the differences between these correlations are small; (3) the correlations fluctuate strongly when measured over different subsets of patients (PMID: 15308542). These observations indicate that the approach and conclusions made by the authors are flawed. It is unsurprising that the authors see an association between the gene set they define as YAP1-regulated genes and outcome in breast cancer patients, and certainly cannot be taken to show a central role for YAP1 target genes in determining patient outcome. They could just as easily have chosen a random gene signature and found a similar association (PMID: 22028643).

Individual gene comparisons are not informative – hundreds of non-YAP-regulated genes will also correlate with survival (PMID: 15308542).

Sample numbers are relatively low for some compared groups, therefore the robustness of conclusions is questionable –irrespective of the problems associated with their signature approach that are outlined above.

The value of the types of analyses that the authors have carried out are either to improve upon already implemented prognostic / theranostic approaches, or as the basis of hypothesis generation that is then tested experimentally. Neither of these approaches have been taken by the authors. As the authors point out in the discussion, there are already several gene expression signatures that are used clinically for the evaluation of breast cancer. Is their YAP signature present in these already defined signatures? Does the YAP signature show advantages over these already established signatures in terms of outcome prediction? Demonstration of superiority of their signature over already implemented ones would be mandatory before publication could be considered, as in the absence of such data the clinical utility of the results in this study are questionable.

Reviewer 2 Report

In the manuscript presented by Gomathi et al., the expression of YAP1 and YAP1 target genes was evaluated in specific subtypes of breast cancer based on public database (METABRIC and TCGA) to explore its prognostic value in breast cancer. The manuscript is well written, and the hypothesis is clear. Specific questions/concerns about the manuscript are presented below:

  1. As we all known, YAP1 acts as an oncogene, but in figure.2 how to explain high YAP1 expression was associated with better survival outcomes in HR positive patients and no significant in other types of breast cancer patients such IDC and TNBC. I think this issue should be indicated somehow in the article.
  2. In result 3.2 genes with Spearman correlation coefficient (rho) values more than 0.20 and significance of less than 0.05 were shortlisted for each cohort. I think that the correlation coefficient should be more than 0.40 (or below -0.40) to say that the correlation is significant.
  3. In figure.5, how to explain the opposite survival outcome of COL12A1 expression in two public database in different breast cancer types of patients. I think this issue should be indicated somehow in the article.
  4. It is better to use another independent breast cancer database to verify the prognostic value of YAP-target such as COL12A1 in BRCA.
  5. Breast cancer cell lines could be used to test the effect of COL12A1 and thus verify the result of this manuscript.
